# Disempowered Warriors: Insights on Psychological Responses of ICU Patients Through a Meta-Ethnography

**DOI:** 10.3390/healthcare13080894

**Published:** 2025-04-13

**Authors:** Elizabeth Kusi-Appiah, Maria Karanikola, Usha Pant, Shaista Meghani, Megan Kennedy, Elizabeth Papathanassoglou

**Affiliations:** 1Faculty of Nursing, University of Alberta, Edmonton Clinic Health Academy (ECHA), 11405-87th Ave., Edmonton, AB T6G 1C9, Canada; kusiappi@ualberta.ca (E.K.-A.); upant@ualberta.ca (U.P.); shaista@ualberta.ca (S.M.); mrkenned@ualberta.ca (M.K.); 2Department of Nursing, School of Health Sciences, Cyprus University of Technology, 15, Vragadinou Str., Limassol 3041, Cyprus; maria.karanikola@cut.ac.cy

**Keywords:** psychological distress, intensive care experience, disempowerment, inclusivity, quality of care

## Abstract

**Objectives:** to systematically examine and synthesize qualitative evidence on adult patients’ psychological distress during an intensive care unit stay to inform development of interventions tailored to their needs. **Method:** We conducted systematic literature searches in CINAHL, MEDLINE, EMBASE, PsycINFO, Scopus, Dissertations and Theses Global, and Google Scholar databases using predefined eligibility criteria. We synthesized primary qualitative research evidence using Noblit and Hare’s meta-ethnographic approach. Reporting was based on the eMERGe framework. The quality of included articles was assessed by the Critical Appraisal Skills Program tool. **Findings:** We identified 31 primary studies from 19 countries. The studies were of moderate to high quality. Data analysis revealed five themes: “*disempowerment*”, “*altered self-identity*” “*fighting*”, “*torment*”, and “*hostile environment*”. One overarching theme, “*the disempowered warrior*”, captured the perpetual tension between the need to fight for their lives and the need to succumb to the care process. Our synthesis discloses that critically ill patients perceive themselves to be in a battle for their lives; while at the same time they may feel helpless and disempowered. **Conclusions:** Our review revealed the tension between the need to fight for one’s life and the sense of powerlessness in the intensive care unit environment. Although participants recognize the important role of healthcare workers, they desired more involvement, collaboration, control, empathy, and empowerment in the care process. These findings can inform approaches to empowering critically ill patients and managing their psychological responses. Care standards must include distress assessment and management that maximize patients’ empowerment and emotional safety with the care process.

## 1. Introduction

For many patients, an intensive care unit (ICU) hospitalization is associated with varied psychological responses to distressing events, including the fear of death, traumatic memories, and facing an uncertain future [1,2,3]. Adverse psychological responses during ICU hospitalization may contribute to mental health impairments after discharge from the ICU, including anxiety, depression, and post-traumatic stress symptoms [4]. Overall, 45% of post-ICU patients experience mental health problems following ICU hospitalization [5]. Mental health problems may also interact with and aggravate other manifestations of the post-ICU care syndrome (PICS), a condition characterized by physical, cognitive, and mental deterioration [6]. Patients’ quality of life may diminish, and the risk of dying increases within the first two years after discharge in patients affected by PICS [7].

Although the psychological and mental impact of stress during ICU hospitalization requires ongoing assessment and management [8,9], there are still no clinical practice guidelines or best practices for the tailored assessment and management of psychological distress in the ICU. Moreover, the role of nurses and bedside healthcare personnel regarding the management of relevant needs remains unclear [10,11]. Despite several qualitative studies focusing on patients’ lived experience of ICU hospitalization, no systematic syntheses of their’ perspectives and experiences of psychological distress exist. Therefore, we undertook a meta-ethnographic synthesis of available qualitative evidence on patients’ experiences and perceptions of psychological distress in the ICU. This work has implications for navigating nurses’ role in managing ICU-related psychological distress and relevant stressors, improving the overall quality of the ICU experience, and ultimately reducing healthcare spending on psychological and mental health-related adverse post-ICU outcomes [12].

### Objective

We aimed to systematically identify and synthesize qualitative evidence on critically ill patients’ experiences of psychological distress during their ICU hospitalization, and its impact on individuals’ perceptions and meanings of self and of their disease.

## 2. Methods

We pursued a meta-ethnographic synthesis of data premised on Noblit and Hare’s seven steps for a cumulative analysis of qualitative data [13]. The interpretive nature of meta-ethnography enables the integration and synthesis of concepts across diverse studies, facilitating a more nuanced theoretical comprehension of ICU experiences by identifying novel insights and generating higher-order constructs. This method was therefore chosen to gain a deeper understanding of the psychological responses and related adaptive mechanisms in ICU patients by translating and synthesizing findings across studies [14].

Our meta-ethnographic approach involved seven steps: defining the research question (Step 1), deciding what was relevant through a systematic database search and a quality appraisal (Step 2), synthesizing included studies and translating articles into one another (Steps 3–6); and reporting findings of the synthesis with the eMERGE framework (Step 7) [14] (Table 1). The review was guided by the following question: “What are adult ICU patients’ experiences, interpretations, and manifestations of psychological distress while in the ICU and its impact on individuals’ perceptions and meanings of self and of their disease?” Below we present the components of the research question using the SPIDER format [15].

### 2.1. Eligibility Criteria

Guided by the SPIDER tool, we included peer-reviewed qualitative studies of empirical data, published in English since 2002, on ICU patients ≥ 18, and addressed the following: (i) psychological responses during ICU hospitalization regardless of when data were gathered (i.e., within or post-ICU), and (ii) patients’ interpretation of psychologically distressing experiences. We excluded reviews, conference papers, abstracts without full text, discussion/opinion papers, and book chapters. Papers that addressed psychological responses of healthcare workers, infants, children, and families were also excluded. We retrieved articles published from 2002 onward, as the first ICU agitation and sedation guidelines were published in 2002 [17], and thus the context of critical care may have been different prior to 2002.

### 2.2. Information Sources and Search Strategy

In collaboration with the research team, an experienced health sciences librarian designed and executed our search strategies for each database. The searches were last updated in August 2024. We searched the following six electronic databases: (1) Medline via OVID, (2) EMBASE via OVID, (3) PsycINFO via OVID, (4) CINAHL via EBSCOhost, (5) Scopus via Elsevier, and (6) Dissertations and Theses Global via ProQuest.

The search strategy for each database was derived from two main concepts; (1) adult ICU patients; and (2) psychological responses, including language for anxiety, fear, and stress disorders such as post-traumatic stress disorder. A combination of natural language keywords and controlled language subject headings were used to construct the search strategy for each database (Table A1, Table A2, Table A3, Table A4, Table A5 and Table A6). We completed forward citations using Scopus and reviewed reference lists of included articles to identify potential articles.

### 2.3. Selection of Sources of Evidence and Data Charting

Retrieved articles were exported into the Covidence software (www.covidence.org) for data management, removal of duplicates, and independent screening. Two team members independently screened titles and abstracts for eligibility, and conflicts were resolved with a third reviewer. Inter-rater reliability was achieved through a mutual agreement on the type of studies to be included.

### 2.4. Quality Assessment

Two authors independently used the Critical Appraisal Skills Program [18] qualitative studies checklist to identify methodological shortcomings of this body of literature and to inform the synthesis and discussion. The CASP qualitative checklist addresses the validity, methodological coherence, and significance of a study.

### 2.5. Data Extraction and Analysis

We developed a data extraction tool based on the review questions, which was iteratively enriched through familiarization with the evidence. Two reviewers independently extracted data to ensure rigor and reliability. Disagreements were resolved through discussion and, when necessary, by consulting with the team to reach a consensus. The analysis team comprised five researchers, including two nursing professors, two Ph.D. students, and one Master of Nursing-prepared nurse. Team members read the included studies and independently developed interpretation themes. The development of high-order interpretation was achieved through team effort. During this phase, the team met virtually, and contributions from team members’ interpretative reading of findings were compared to ascertain the meaning of the conceptual data in a high-level interpretation (third-order constructs). The emerging themes were confirmed through consensus. In meta-ethnography, the concept of saturation is adapted to the context of synthesizing existing research rather than collecting new data. In our study, saturation was used to ensure conceptual clarity, that is, that the synthesis addressed the research questions adequately and represented the breadth and depth of available research data, while additional analysis did not contribute to the emerging themes [13,14].

## 3. Results

A total of 4862 articles were screened, out of which 4775 were excluded. We assessed 87 full-text articles based on the eligibility criteria and 31 studies met the inclusion criteria (Figure 1: PRISMA chart).

### 3.1. Study and Participants’ Characteristics

A total of 31 studies from 19 countries were synthesized (Figure A1). Table 2 provides a summary of the selected studies. The results of second- and third-order analysis (Table A7). A wide range of methodological and theoretical perspectives were employed in the reviewed studies (Figure A2). Most of the studies (n = 21) were conducted in medical and surgical ICUs (Figure A3). The included studies varied in terms of ICU context across countries, including factors such as ICU design, nurse-to-patient ratios, and the standardized ICU protocols/treatment strategies. However, no studies reported information on the ICU context. Participants included in the primary studies were aged between 20 and 88 years. In most studies, the majority of participants were male (Figure A4), with no neurocognitive or psychiatric limitations, and mechanically ventilated during their ICU stay (Table 2 and Figure A5). The psychiatric history and cognitive status of participants were unreported in most studies [2,3,19,20,21,22,23,24,25,26,27,28,29]. Some participants experienced delusions, delirium, and/ or hallucination during ICU hospitalization [23,26,29,30,31,32,33,34,35].

### 3.2. Quality Appraisal

The studies were of moderate to high quality; yet some did not adequately disclose the researchers’ relationship with participants or address the potential influence of researchers’ bias and positionality on data collection and interpretation (eight studies) (Table 3).

### 3.3. Experiences of the Participants in the Reviewed Studies

Data analysis revealed five themes: “disempowerment”, “altered self-identity” “fighting”, “torment” (with subthemes “trauma” and “fear of death”), and “hostile environment” (Figure 2). One overarching theme, “the disempowered warrior”, emerged from the culmination of these themes and captured the perpetual tension between the need to fight for their lives and the need to succumb to the care process. The themes with participants’ quotes are illustrated in Table 4.

#### The Disempowered Warrior

The overarching theme, “*the disempowered warrior*”, captures participants’ perceptions of the simultaneous urgency to fight for their survival and an overwhelming sense of helplessness. The ICU care equipment and process of care seemed to threaten their sense of stability, safety, and personal identity. Although participants described their struggle to return to their pre-ICU state, they felt that the ICU system disempowered and dehumanized them, making it difficult to effectively confront the challenges of critical illness and its care. Although they perceived themselves to be in a constant battle for survival, their agency over their fate was unacknowledged: “*You become very anxious, it is like you are fighting for your life, … you are just getting instructions push, breath or whatever…*” [19]. The themes that culminated into this overarching theme are described below:

##### Disempowerment

Disempowerment was predominantly linked with participants’ perceptions of being deprived of their rights and experiencing dehumanization: *“[…] I’m not an engine that needs to be fixed, I mean, I’m a human being*” [36]. Many described that they lost control [25,34,37]: “*Being in the ICU is just like a little child who could not swim [and] was thrown into a cold swimming pool*” [29] and perceived that they could not influence care decisions [25]: *you know, not giving updates […] you just struggle. You feel so helpless […]*” [34].

Furthermore, participants described a loss of autonomy in several instances, including the right to information, to keep personal or religious items, to adequate physical, psychological, and spiritual care, and to dignity [3,19,24,25,38,39]. Frustration due to experiences of power imbalance was evident: “*when they put this and that on me, I was very annoyed, but I must give in. Being in the status as a patient makes [patients] passive, meaning we have to put utmost trust on doctors and nurses. Our lives are in their hands*” [24]. Some participants wanted to challenge uncomfortable care processes or influence care decisions but felt powerless to do so. They perceived that they were helpless in diseased bodies [25,38]. Others felt “*like a vegetable…. very defenseless and helpless*” [40]. Nevertheless, participants desired to be acknowledged as humans by healthcare workers [12,35,36,41]. Specifically, participants sometimes perceived staff attitudes as neglectful rather than indifferent [3]: “*You… [ring] the bell and no one shows up. […] It made me feel helpless, and powerless. … what can I say, you feel hurt and you could easily start to cry*” [36]. Participants also felt that their concerns were invalidated, dismissed, or not taken seriously, further undermining their dignity [25,36]: “*You want to be treated as an adult instead of a child, right? […]*” [36]. Across studies, participants described not receiving timely explanations or updates about their condition, care plan, or prognosis [20,25,26,29,36], as one participant explained: “*you know, not giving …updates … you just struggle. You feel so helpless, and you wonder if you are going to die*” [34]. This lack of information often led to death anxiety, delusions, and sometimes paranoia about healthcare staff’s intentions [31]. For instance, in Wang et al.’s (2009) study, the following was shared: “*someone pressed my arms and legs forcefully, some others opened my mouth and put something into my throat, after that I knew that was the tracheal tube, why did nobody tell me about it before that?*” [29].

Regarding dehumanization, participants described vivid experiences of depersonalization, and their desire to be understood and recognized as human beings [12,35,36,41]. McKinley et al., (2002) noted that depersonalization occurred when staff treated patients as objects, or talked about them as though or not they were not present, especially during ward rounds [26]. Overpowering noise from staff conversations or laughter further contributed to the perception of impersonal care [27,35,36,41]. Participants recalled the following: “*we were lying on bed like vegetables, it seemed that we did not exist to the nurses, they were chatting and laughing, too noisy*” [29], or “*There were moments when [patients] almost cried in despair. [They were] suffering and [care practitioners] laugh[ed] at the top of their lungs a few meters away*” [33].

##### Altered Self-Identity

Participants described significant alterations in their perceptions of self, body, and competencies, with the loss of self-care abilities being central to this altered perception of self-identity [29,35]. Many participants felt vulnerable due to extreme physical and emotional dependency on care providers, medical equipment, and technologies [1,24,25,26,29,33,34,35,42]. This reliance on technology led them to feel as though their physical existence was controlled by machines [24,25], sparking insecurities about losing their sense of self [24,36]. Specifically, dependence on life-sustaining technology affected their perception of their bodies, which they described as “freakish” and incapable of performing normal functions [3,20,24,37]: “*Weak in [the] body … [some] felt like a child, … who doesn’t think too much, can’t do anything; can’t pee by … self, poo by … self, can’t do anything*” [25].

The loss of functional independence led participants to worry about whether they would ever regain it [3,20,22,23,31,41]: “when I woke up from the coma the only thing that I confronted was fear, i.e., if I would be able to speak again because of the tracheotomy and if I would be able to walk again because I was confined in my bed…” [3].

Many studies underlined feelings of embarrassment and loss of dignity due to the inability to perform self-care functions, such as toileting, eating, and drinking, [19,21,26,33,38] as a loss of personal dignity [19,33]. Ashkenazy et al. (2021) interestingly noted that the level of embarrassment varied according to participants’ beliefs and values [38].

Perceptual distortions, including hallucinations and delusions, further affected participants’ sense of time, space, and interactions with care providers [3,21,27,28,29,31,33,35,36,40,41]. Some participants described themselves as becoming introspective or paranoid whilst in the ICU [3]. The lack of social connections, which were deemed as essential for nurturing their identity, further deepened these distressing experiences [19,20,21,27,28,34,37,38,40].

##### Fighting

The notions of battle and fighting for survival were evident throughout participants’ narratives, and they underscored their perceptions of trauma and fear of death. Participants fought not only for their survival, but also to be heard and understood by the care team and to maintain their psychological equilibrium.

When healthcare providers focus solely on implementing care routines and achieving predefined healthcare goals, patients often perceive the experience as distressing and oppressive, intensifying their fight for survival [3,36,40,43]. This exclusive focus on physical interventions can overshadow the need for clear communication and emotional support, leaving patients feeling fearful, abandoned, and misunderstood. One patient expressed the sense of neglect, stating, “*I am sure that they want to hurt me, and nobody explains to me what has happened. The only thing that they cared about is my coughing*” [3]. These experiences highlight the centrality of psychological well-being in the fight for survival. Another participant described experiences of anxiety, likening it to a constant battle for life: “*You become very anxious, it is like you are fighting for your life, … you are just getting instructions push, breath or whatever…*” [19]. Patient narratives emphasized the need for compassionate care that addresses both the physical and emotional dimensions of patient experiences [19,43]. Without such support, the ICU becomes not just a place of medical intervention, but a site of profound psychological distress, intensifying the patient’s trauma. The profound disorientation and helplessness of the ICU environment are captured in the analogy, “*Being in the ICU is just like a little child who could not swim [and] was thrown into a cold swimming pool*” [29]. Additionally, patients perceived lack of support in their fight for their lives. A patient detailed how emergency procedures in the ICU worsened personal distressing experiences: “*when a patient becomes ill, the ward falls apart, all the doctors and nurses are gone [leaving other patients] subconsciously restless and upset*” [2].

##### Torment

Admission to the ICU inflicted suffering and feelings of imprisonment, as noted in five studies from Jordan, Iran, Italy, China, and Greece [3,12,29,39,42].

Participants used expressions such as horrible, martyrdom, entombed, jail, and hell [1,3,29,34,42] to communicate their ICU experiences: “*Like a prisoner with tied hands and feet and closed mouth who is continually punctured with needles and cannot say anything*” [42]. Thus, care routines were often conceptualized as torturous and impersonal [3,19,26,37,42]. For instance, experiencing immobility, instrumentation, intubation, painful medical procedures, or being restrained in the ICU translated into feelings of terror, confinement, agitation, and isolation [1,3,34,41,42]. Indeed, participants often shared an overwhelming feeling of loneliness: “*an infinite sense of loneliness as the most negative feeling*” [3], describing that that loneliness could make “you die in bed” [19]. Furthermore, loneliness was exacerbated when care providers focused more on technology than on the participants, underscoring their need for greater connection with people around them [27,28,37,38,39].

Discomfort from intubation and extubation were also described as oppressive, and most participants could not withstand the emotional and physical discomfort of endotracheal suctioning [12,19,20,21,27,28,34,35,40,42]. Participants “*waited for this torment to end” (Zisopoulos et al., 2022), as “it was one of those things that you have to kind of grin and bear it*” [3].

Trauma

Participants interpreted their illness and care context as extremely stressful and traumatic, not only for themselves, but also for their spouse, children, and loved ones [12,22,27,34,41,43]. They reported a myriad of emotional responses: fear, worry, anxiety, sadness, sorrow, despair, disappointments, grief, aversion, loneliness, and insecurity [3,12,19,20,21,22,25,27,28,29,34,35,36,37,40,42,43,44]. In a study in Iran, the experience of waking up in an ICU was described as tremendously terrifying: “*to be honest, everyone who wakes up and does not know where he is and once sees all these devices hooked to him and cannot do anything, is definitely going to have another MI [myocardial infarction] from fear*” [20].

Participants worried about their prognosis and risk of disability [12,22,27,34,39,41,43]. They also feared developing complications such as mouth ulcers or breathing problems from invasive respiratory procedures [12,19,20,21,22,25,27,28,29,34,36,37,40,41,42,43]. Participants sometimes received contradictory statements about their conditions and hopes for treatment which made them more confused and vulnerable [2,3,19]. “*Today, the doctor comes and says that the operation was very good, you will get up soon and you will not have any special complications. Tomorrow another doctor will come and say that it is too early for me to comment. Maybe you will have a series of complications after the operation!*” [2]. These experiences were not different from what was expressed in a Greek and a South African study: “*There is a lot of can’t… you can’t do …so all of these accumulate to a bad experience”* [19]; “*From the moment you woke up you wanted to get out, to flee, you don’t want to stay even a minute longer*” [3].

Fear of Death

Participants associated admission to ICU with the risk of death: “I was very scared when I realized I was in the ICU. I told myself I was done!” [2]. Another participant described: ‘when I woke up I found a tube in my throat, […] I could not stop thinking: Was I dying soon?” [29]. These narratives were echoed in most participants’ stories [1,27,28,35,37,39,42]. Reports of death anxiety also stemmed from witnessing or hearing about the death of other patients “We were four and I was the only one left. Three died, and I saw them die…It was not easy holding on to the bed waiting for your moment to come” [20,33,37]. Some patients willfully desired to die to end their suffering [24,42]. One participant explained, “I have no fear of dying, but always of being in pain and suffering …. the same pain, I would like to die instead. … I thought that if (I) didn’t die soon, it would be pitiful” [24].

##### Hostile Environment

Participants perceived the care environment as hostile and traumatic, and they felt the need to fight for their lives, and, at the same time, to resist perceived harm. The ICU was also described as “*an artificial environment, inappropriate to the human being, [and] not compatible with human expectations*” [33]. The participants also experienced exhaustion and reduced quality of life from the ICU environment. Sounds and lights were described as tormenting [1,3,12,25,26,27,29,36,37,38,40,41]. They described that “*for almost the entire [time], all the lights were on, there was no window, no clock, and [patients] didn’t have a sense of time*” [3] and in effect contributed to the hostility of the environment [35,37,41]. Consequently, some participants became tired and wary of their situation: “*can hear all of it… it’s hard to explain but if you can’t sleep, can’t shut yourself down so to speak, and being afraid of all that may happen makes you scared. Yes all that makes you feel… I can’t hang on anymore*” [36].

Technology was perceived as both maintaining their life and a constant source of anxiety and fear, as participants tried to interpret if the cues from the machines surrounding them signaled threats. Participants also dreaded experiencing technology failure or being harmed by technology [20,24,25,37]: “*I kept thinking of this: What should I do if there was anything wrong in the machine without someone knowing? Was that my last day of life?*” [36]. As their stress was perpetuated, some participants demonstrated a form of resistance to care [19,23,29,36,38]. For instance, Gilder et al., (2022) [23] noted a case of a patient being panicky, biting the endotracheal tube and trying to pull the tube out. Wang et al., (2009) [29] described that some participants were resistant to the equipment and technologies used for treatment due to the discomfort and suffering that came with them. One patient shared: “*I did not know what is going on, I woke up, I felt a tube, I thought maybe I will pull a little bit, suddenly I saw the tube halfway out, so I took it out*” [38].

**Table 2 healthcare-13-00894-t002:** Summary of included studies.

Authors and Country	Aim	Sample Characteristics	Themes
Adeyemi (2016) [19] South Africa	To explore and describe the experiences of patients on mechanical ventilation in the intensive care unit of one public sector hospital in Johannesburg.	Female = 4, male = 6. Age = 39–48 years. Average length of stay in ICU (LOS): 4.9 days. Cognitive status: not reported. Mean duration on mechanical ventilation = 57.45 h. Sedation status: all patients were sedated.	Experience of fear Feelings of bothering Powerlessness Sense of safe/unsafe Body image Spirituality
Albanesi et al. (2020) [12] Italy	To describe the lived experiences of patients who underwent cardiac surgery.	Female = 9, male = 2. Age = 42–75 years. Cognitive status: consciousness. Ventilation status: not reported Sedation status: One patient described being on sedation.	Endless time in ICU “trapped”, “in prison”, and “not understanding time” Anchor in the storm Presence of nurses, sense of security, and trust
Alpers et al., (2012) [43] Norway	To gain knowledge on what factors contribute to inner strength in critically ill patients cared for in an intensive care unit.	Female= 3, male =3. Age = 60–72. Ventilation and sedation status not reported.	To have the support of next of kin The wish to go on living To be seen
Ashkenazy et al., (2021) [38] Israel	To describe mechanically ventilated ICU patients’ perception of discomfort and how they differentiate discomfort from pain.	1 males, 3 females. Age 19–81 years. Ventilated for 2–8 days. LOS 3–20 days. Participants were fully conscious, oriented, and able to articulate their ICU experience. Most interviews were conducted 3–8 days post ICU discharge.	Unpleasant physical sensations Unpleasant psychological feelings: Embarrassment, helplessness, loneliness, and fear
Aslani et al., (2017) [20] Iran	To explore the psychological experiences of patients under mechanical ventilation following an open-heart surgery in 2016.	Female = 8, male = 7. age range = 40–60. Cognitive status: not reported. All patients had received ventilation. Sedation status: all patients were sedated.	Anxiety, fear, despair, and dependency Spiritual connection Presence of treatment team, communication Family (presence, praying)
Engström et al. (2012) [21] Sweden	To describe the intensive care unit experiences of people undergoing mechanical ventilation.	Female = 4, male = 4. Age = 45 to 72 years. Ventilation status: 5–21 days. Sedation status: all patients were sedated. Duration on sedation not specified.	Feeling vulnerable and dependent Struggling to be able to communicate Feeling safe with the staff Appreciation of the diary and follow-up visit
Eqylan et al., (2022) [39] Jordan	To explore critically ill Muslim patients’ experiences and perceptions related to confinement to isolation rooms in Jordan.	5 male and 5 female. Mean age: 42.2 ± 17.3 years. Mean duration of isolation stay: 6.6 ± 2.2 days. Included only patients not ventilated or sedated.	Feeling isolated and imprisoned Losing basic patients’ rights Feeling rejected by healthcare providers Accepting isolation and its adversity
Flahault et al., (2021) [30] France	To capture the intimate experience of ICU patients, to understand how they make sense of this experience, and to explore their experience and representations of an ICU diary.	3 men and 2 women. Age: 24–72 years. LOS: 8–13 days. Duration of mechanical ventilation: 4–9 days. Patients experienced delirium and hallucinations during admission.	The nightmare of the ICU experience The positive image of healthcare workers during intensive care
Foster et al., (2010) [22] United Kingdom	To describe the experience of having a tracheostomy tube as lived by a group of people who had this tube inserted as a part of their critical illness or as a procedure.	Female = 1, male = 2. LOS ≥ 14 days. Cognitive and mechanical ventilation status not reported.	Necessity of communication Retaining normality Psychosocial discomfort Painful procedures Relationships with staff Fear of the unknown
Gilder et al., (2021) [23] New Zealand	To describe the patient experience of the endotracheal tube and suction, following mechanical ventilation in post-operative cardiac surgical patients.	Eight male, two female. Age: 26–84, mean= 64.1 years. LOS: mean = 24.5 h, range= 17–72 h. Duration of mechanical ventilation: mean = 6.3 h range = 4.1–17.4 h.	Hallucinations and the effects of drugs Feeling good about recovery Slow passage of time and sleeping issues Anxiety and concerns about the future
Herbst and Drenth, (2012) [32] South Africa	To increase insight into the thoughts, feelings, and bio-psychosocial needs of the patient receiving treatment in the ICU.	Single participant 27 years old female. Participant regained consciousness. after three weeks in the ICU. Ventilated and sedated but number of days involved not reported. LOS: not reported.	Helplessness and inability Mortality and injury Loneliness and isolation Specific emotions Chaos and delusion Social support systems Hopeful messages
Hofhius et al., (2008) [36] The Netherlands.	To evaluate the perceptions of patients regarding nursing care in the ICU, and secondly, to explore patients’ perceptions and experiences of their ICU stay.	Female= 4, male =7. Age = 54–73 years. Ventilation status: 4–20 days. Sedation status: all patients were sedated.	Providing the seriously ill patient with information and explanation Personal approach by the nurse Placing the patient in a central position
Karlsson et al., (2011) [25] Sweden	To illuminate the lived experience of patients who were conscious during mechanical ventilation in an intensive care unit (ICU).	Female = 3, male = 9. Age = 23–88 years. LOS = 2–23 days. Ventilation status: All patients were ventilated. Number of days on ventilation not stated. Motor activity assessment scale = 3–4. Cognitive status: not reported. Sedation status: no patient was under sedation during interview.	Being dependent on mechanical ventilation to survive Being forced to submit to the will of others Having to submit to other people’s willingness to understand non-verbal communication Being out of control Having to endure Experiencing a sense of control Yearning for independence Comprehensive understanding Being viewed as a participant and companion
Locsin and Kongsuwan, (2013) [24] Thailand	The purpose of this study was to describe the meaning of the experiences of patients who were dependent on technologies while being cared for in ICUs.	7 male, 3 female. Age: mean = 43.9 years, range = 22–76 years. LOS: mean = 50.8, range = 2 days–3 months. Ventilation status: patients were ventilated but number of days on ventilation noted reported.	Fear and insecurity of not being one’s self Nightmare Death is better than suffering Loss of autonomy Sustaining life through family support Not wanting to burden the others Being in trust and security Transitioning to a better life Discovering new meanings of living
McKinley et al., (2002) [26] Australia	To gain an understanding of the experience of being a seriously ill patient in an (ICU).	8 males, 6 females. Aged 17–71. LOS: 3–53 days. Ventilation status: not reported.	Vulnerability Security Salutary experiences Difficulty communicating and lack of knowledge Frightening physical experiences Lack of sleep Cognitive changes Pain and discomfort Personalizing care Needs anticipated and met
Mortensen et al., (2023) [31] Denmark	To explore everyday life experiences of critically ill patients with delirium during the ICU stay, from ICU discharge until 1-year follow-up, focusing on their health-related quality of life and cognitive function.	9 women and 8 men. Age: median = 69 years, range = 57–73.5. LOS: median= 8, IQR = 5.5–26.5. Ventilation status: not reported.	Struggling to regain a functional life Struggling to regain normal cognition Distressing manifestations from the ICU
Mylén et al., (2016) [41] Sweden	To explore the lived experiences of conscious patients in neurosurgical intensive care.	Female = 7, male = 4. Age range = 37–81. LOS = 5–15 days. Cognitive status: GSC 14—15. Ventilation status: no patients were ventilated.	The essence: To feel strong in an unfamiliar situation To feel safe in an unfamiliar situation To experience strains and limitations To be confirmed as a human being
Olausson et al., (2013) [37] Sweden	To reveal the meanings of the ICU settings as a place of care.	Female = 5, male = 4. Age range = 38–52. LOS = 5–90 days. Ventilation status: not reported. sedation status: not reported.	A life-affirming place A place of tenderness and care A place in-between A place of trust and security An embodied place
Olsen et al., (2017) [35] Norway	To investigate how adult ICU patients experienced the ICU stay, their recovery period, and the usefulness of the pamphlet.	Female = 10, male = 19. Age range = 20–80. Ventilation status: ventilated for ≥ 48. Sedation status: all patients were sedated.	Floating between facts and delusions To understand and to be understood Valuing family
Pakmehr et al., (2017) [42] Iran	To phenomenologically explore the intubated patients’ lived experiences of ICU care.	Female = 7, male = 5. Age = 19–48. Ventilation = 2–50 days. Sedation status: not reported.	Mental health problems associated with the lack of communication Family as a need for the patients Intubation as an evocation of imprisonment Torture and death
Roberts et al., (2019) [34] United States	To determine the pervasive patient experience components and how the experience may have been influenced by targeted, light sedation practices.	Female = 5, male = 10. Average age = 61.8 Ventilation status: for less than 14 days. Average LOS = 3.5 days. Sedation status: Richmond agitation Sedation scale score = 1.7.	Communication difficulties Feelings of discomfort Positive interaction with people Inattentive staff Unclear recollections
Rodriguez-Almagro et al., (2019) [27] Spain	To explore the perceptions about the experiences of patients in the ICU, their family members, and the nurses who attend them.	Female = 4, male = 5. Age = 22–67. Cognitive status: not reported. ventilation status: not reported.	The experience of humanization and dehumanization Fear Feeling of suffering related to noise and pain
Samuelson et al., (2011) [40] Sweden	To describe unpleasant and pleasant memories of the ICU stay in adult mechanically ventilated patients.	Female = 122, male = 128. Age = 63.4 ± 13.8. Cognitive status: no confusion, delirium. Median LOS = 3.4. Ventilation status: more than 24 h. Sedation status: not reported.	Acute existential distress Emotional agony Agitation Depressive feelings Loss of control
Sanson et al., (2021) [33] Italy	To explore the experiences of being cared for in an intensive care unit (ICU) through patients’ memories.	Female 42; male 58. Age: 66 (53–77) years. Glasgow Coma Scale: 13 (10–15). Duration of ventilation: 23.5 h (2–67). LOS: 69 h (40–141).	Discomfort Loss of dignity Mood fluctuations: feeling anxious, frightened, closeness to death, and worry about relatives Unheard requests Delusional memories
Schou and Egerod, (2009) [28] Denmark	To provide a contemporary description of the patient experience of weaning.	Female = 2, male = 8. Age range = 35–84. Cognitive status: not reported. Ventilation status: ≥24 h.	Discomfort and impaired communication Loss of control and loneliness Existential phenomena Temporality and human interactions
Takashima et al., (2018) [1] Japan	To clarify the stress experiences of patients mechanically ventilated in an intensive care unit (ICU) for more than 12 h.	22 women and 74 men. Average age = 69.4± 11.5. Average duration of intubation: 54.7 ± 60.0 hr. Average LOS: 6.8 ± 5.9 days.	Unbearable holistic discomfort. Pain of being unable to control self.
Tavakoli et al., (2022) [2] Iran	To study the experiences of discomfort and self-management strategies in patients admitted to the ICU.	5 females and 8 males. Age range = 17–61 years. LOS: not reported. Ventilation status: not reported. Cognitive status: conscious for at least 72 h after admission to the ICU.	Fear of disability and possible death Separation from family Understanding ambiguity and contradiction in treatment Environmental disruptors Painful and unfamiliar devices and treatments Recourse to spirituality Benefiting from psychosocial coping
Vogel et al., (2021) [44] Sweden	To explore patients’ patterns of behaviour during the process from becoming critically ill to recovery at home.	4 men and 9 women. Age = 22–82 years. LOS = 2–28 days. Ventilation status: not reported.	Recapturing life Recoding life Emotional balancing
Wang et al., (2009) [29] China	To understand patients’ intensive care experience while receiving mechanical ventilation in intensive care units.	Female = 3, male = 8. Age = 33–78 years. Mean LOS= 174.6 h. Cognitive status: not reported. Mean length of mechanical ventilation = 107.3 h. Sedation status: all patients were sedated.	Being in an unconventional environment Physical suffering Psychological suffering
Yahui et al., (2022) [45] China	To explore the experience of ICU patients and their relatives and to investigate their daily needs within the closed management system in many Chinese ICUs.	6 male and 9 female. Mean age: 59.60 ± 17.33, range = 28~88 years. Cognitive status: able to communicate verbally. LOS: at least 72 h. Ventilation status: not reported.	Variety of feelings Staff assistance required Expectations for relatives
Zisopoulos et al., (2022) [3] Greece	To describe how participants describe their ICU experiences and make sense of their experience.	17 males, 9 females. Age = 20–68 years. LOS = 2 weeks. Ventilation status: not recorded. Cognitive status: unclear.	The martyrdom in the ICU: a place where you are buried Difficult moment touched by death Return from the utter void Efforts to regain touch with reality The essential role of family and healthcare professionals

**Table 3 healthcare-13-00894-t003:** Quality Appraisal with CASP Tool.

Included Studies	Was There a Clear Statement of the Aims of the Research?	Is a Qualitative Methodology Appropriate?	Was the Research Design Appropriate to Address the Aims of the Research?	Was Recruitment Strategy Appropriate to the Aims	Were Data Collected in a Way that Addressed the Research Issue?	Has the Relationship Between Researcher and Participants Been Adequately Considered	Have Ethical Issues been Taken into Consideration?	Were the Data Analysis Sufficiently Rigorous?	Is There a Clear Statement of Findings?	How Valuable Is the Research?	Overall Risk of Bias Assessment
Albanesi et al. 2020 [12]	Yes	Yes	Yes	Yes	Yes	No	Yes	Yes	Yes	Greatly	low
Ashkenazy et al. 2021 [38]	Yes	Yes	Yes	Yes	Yes	Yes	Yes	Yes	Yes	Greatly	low
Aslani et al., 2017 [20]	Yes	Yes	Yes	Yes	Yes	No	Yes	Yes	Yes	Moderately	moderate
Hofhuis et al., 2008 [36]	Yes	Yes	Yes	Yes	Yes	No	Yes	Yes	Yes	Moderately	moderate
Olsen et al., 2017 [35]	Yes	Yes	Yes	Yes	Yes	No	Yes	Yes	Yes	Moderately	moderate
Rodriguez-Almagro et al., 2019 [27]	Yes	Yes	Yes	Yes	Yes	No	Yes	Yes	Yes	Greatly	low
Alpers et al., 2012 [43]	Yes	Yes	Yes	Yes	Yes	Yes	Yes	Cannot tell	Yes	Moderately	moderate
Engström et al. 2012 [21]	Yes	Yes	Yes	Yes	Yes	No	Yes	Yes	Yes	Moderately	moderate
Samuelson et al. 2011 [40]	Yes	Cannot tell	Yes	No	No	No	Yes	Cannot tell	Yes	Moderately	moderate
Karlsson et al., 2011 [25]	Yes	Yes	Yes	Yes	Yes	No	Yes	Yes	Yes	Greatly	low
Wang et al., 2009 [29]	Yes	Yes	Yes	Yes	Yes	Yes	Yes	Yes	Yes	Moderately	low
Olausson et al., 2013 [37]	Yes	Yes	Yes	Yes	Yes	No	Yes	Yes	Yes	Moderately	moderate
McKinley et al., 2002 [26]	Yes	Yes	Yes	Yes	Yes	No	Yes	Yes	Yes	Greatly	low
Adeyemi, 2016 [19]	Yes	Yes	Yes	Yes	Yes	No	Yes	Yes	Yes	Greatly	low
Pakmehr et al., 2017 [42]	Yes	Yes	Yes	Yes	Yes	No	Yes	Yes	Yes	Moderately	moderate
Roberts et al., 2019 [34]	Yes	Yes	Yes	Yes	Yes	No	Yes	Yes	Yes	Greatly	low
Schou and Egerod, 2009 [28]	Yes	Yes	Yes	Yes	Yes	Yes	Yes	Yes	Yes	Moderately	low
Foster, 2009 [22]	Yes	Yes	Yes	Yes	Yes	Yes	Yes	Yes	Yes	Greatly	low
Mylen et al. 2015 [41]	Yes	Yes	Yes	Yes	Yes	No	Yes	Yes	Yes	Moderately	moderate
Vogel et al., 2021 [44]	Yes	Yes	Yes	Yes	Yes	No	Yes	Yes	Yes	Moderately	moderate
Tavakoli et al., 2022 [2]	Yes	Yes	Yes	Yes	Yes	No	Yes	Yes	Yes	Greatly	low
Zisopulos et al. 2022 [3]	Yes	Yes	Yes	Yes	Yes	No	Yes	Yes	Yes	Greatly	low
Locsin and Kongsuwan, 2013 [24]	Yes	Yes	Yes	Yes	Yes	No	Yes	Yes	Yes	Greatly	low
Herbst and Drenth, 2012 [32]	Yes	Yes	Yes	Yes	Yes	No	Yes	Yes	Yes	Moderately	moderate
Mortensen et al. 2023 [31]	Yes	Yes	Yes	Yes	Yes	Yes	Yes	Yes	Yes	Greatly	low
Flahault et al., 2021 [30]	Yes	Yes	Yes	Yes	Yes	Yes	Yes	Yes	Yes	Greatly	low
Gilder et al., 2022 [23]	Yes	Yes	Yes	Yes	Yes	No	Yes	Yes	Yes	Greatly	low
Sanson et al. 2021 [33]	Yes	Yes	Yes	Yes	Yes	No	Yes	Yes	Yes	Greatly	low
Eqylan et al., 2022 [39]	Yes	Yes	Yes	Yes	Yes	No	Yes	Yes	Yes	Greatly	low
Yahui et al., 2022 [45]	Yes	Yes	Yes	Yes	Yes	No	Yes	Yes	Yes	Greatly	low

**Table 4 healthcare-13-00894-t004:** Themes and sub-themes.

Participant Quotes	Main Theme
*“…I couldn’t get to sleep once, but that battle Axe [nurse] said I had*” [25]. “*You want to be treated as an adult instead of a child, right?*” [36] “*Yes, I mean I’m not an engine that needs to be fixed, I mean, I’m a human being*” [36] “*you know, not giving …updates … you just struggle. You feel so helpless, and you wonder if you are going to die*” [34] “*we were lying on bed like vegetables, it seemed that we did not exist to the nurses, they were chatting and laughing, too noisy*” [29] “*There were moments when [patients] almost cried in despair. [They were] suffering and [care practitioners] laugh[ed] at the top of their lungs a few meters away*” [33] “*You… [ring] the bell and no one shows up. […] It made me feel helpless, and powerless. … what can I say, you feel hurt and you could easily start to cry.*” [36] “*someone pressed my arms and legs forcefully, some others opened my mouth and put something into my throat, after that I knew that was the tracheal tube, why did nobody tell me about it before that?*” [29]	Disempowerment
*“Weak in [the] body……[some] felt like a child, … who doesn’t think too much, can’t do anything; can’t pee by…self, poo by …self, can’t do anything*” [25] “*like a vegetable… very defenseless and helpless*” [40] “*when I woke up from the coma the only thing that I confronted was fear,* i.e.*, if I would be able to speak again because of the tracheotomy and if I would be able to walk again because I was confined in my bed…*” [3] “*felt strange… like there was a gap between [the] body and mind*” [1] *“It felt like my upper body was like an elephant, I didn’t expect my legs could carry me*” [41] “*I didn’t know what they were doing to me…I was a body, there, available*” [33]	Altered self-identity: altered self perception, self-image and self competency
“*Being in the ICU is just like a little child who could not swim [and] was thrown into a cold swimming pool* [29] “*You become very anxious, it is like you are fighting for your life, … you are just getting instructions push, breath or whatever…*” [19] “*I am sure that they want to hurt me and nobody explains to me what has happened. The only thing that they cared about is my coughing.*” [3] “*when a patient becomes ill, the ward falls apart, all the doctors and nurses are gone [leaving other patients] subconsciously restless and upset*” [2]	Fighting
“*Like a prisoner with tied hands and feet and closed mouth who is continually punctured with needles and cannot say anything*” [42] “*you do not see anyone, all in bed [with] a series of devices …connected to [them that] make noise*” [2] “*You [can] not move because of the tubes... you are attached to the bed*” [38] “*Since you are intubated, they passed it through the nose and the nose got sore. When the sore was recovering, they pulled it out and inserted [nasogastric tube] again into the nose. Doing this, they pressed the injured sore which made more pain*” [42] “*It was one of those things that you have to kind of grin and bear it*” [34]	Torment
“*to be honest, everyone who wakes up and does not know where he is and once sees all these devices hooked to him and cannot do anything, is definitely going to have another MI [myocardial infarction] from fear*” [20] “*Today, the doctor comes and says that the operation was very good, you will get up soon and you will not have any special complications. Tomorrow another doctor will come and say that it is too early for me to comment. Maybe you will have a series of complications after the operation!*” [2] “*There is a lot of can’t… you can’t do …so all of these accumulate to a bad experience*” [19] “*From the moment you woke up you wanted to get out, to flee, you don’t want to stay even a minute longer*” [3]	Trauma
“*I was very scared when I realized I was in the ICU. I told myself I was done!*” [2] “*when I woke up I found a tube in my throat, […] I could not stop thinking: Was I dying soon?*” [29] “*We were four and I was the only one left. Three died, and I saw them die… It was not easy holding on to the bed waiting for your moment to come*” [20,33,37] “*I have no fear of dying, but always of being in pain and suffering… the same pain, I would like to die instead. … I thought that if (I) didn’t die soon, it would be pitiful* [24] *I was in fear. I didn’t want to take my eyes off from the monitors…When the machine alarmed, I was frightened. Why did it alarm? What was happening with me?*” [24] “*I kept thinking of this: What should I do if there was anything wrong in the machine without someone knowing? Was that my last day of life?*” [29] “*I was scared because I thought the night nurse will kill me…so I yelled at her…*” [31]	Fear of death
“*for almost the entire [time], all the lights were on, there was no window, no clock, and [patients] didn’t have a sense of time*” [3] “*can hear all of it… it’s hard to explain but if you can’t sleep, can’t shut yourself down so to speak, and being afraid of all that may happen makes you scared. Yes all that makes you feel… I can’t hang on anymore*” [36] “*I did not know what is going on, I woke up, I felt a tube, I thought maybe I will pull a little bit, suddenly I saw the tube halfway out, so I took it out*” [38] “*when a patient becomes ill, the ward falls apart, all the doctors and nurses are gone [leaving other patients] subconsciously restless and upset*” [2] “*This thing that I put my wrist (referring to manometer connected to an arterial line) makes me both annoyed and afraid to make the slightest moan, because I think it will break and be dangerous for me!*” [33] “*an artificial environment, inappropriate to the human being, [and] not compatible with human expectations*” [33]	Hostile environment

## 4. Discussion

This meta-ethnographic review synthesized qualitative evidence on the psychological distress of those hospitalized in the ICU. Understanding critically ill patients’ psychological distress can inform the development of measures to assess and manage psychological stressors in the ICU, focusing on patient education, rehabilitation, and evidence-based treatment for ICU-related psychological trauma.

The evidence, drawn from participants’ accounts across diverse cultural contexts, highlights the profound psychological distress linked to the critical care experience. As the majority of the identified studies involved predominantly mechanically ventilated male participants, the generalizability of these findings across different ICU populations is uncertain. Female patients in the ICU often experience significant psychological distress, including higher levels of anxiety, depression, and post-traumatic stress, compared to their male counterparts [46], and targeted studies are needed to elucidate the differences in the experience of ICU psychological distress between men and women.

The central finding of this review is that critically ill individuals consistently strive to maintain a sense of agency, even in their most vulnerable moments. Participants described actively fighting for their lives and preserving their bodily and psychological integrity. This reframes the traditional view of healthcare workers fighting solely on behalf of patients [47], instead highlighting the importance of recognizing patients’ agency and honoring their wishes. This perspective calls for care that prioritizes inclusivity, empowerment, and active partnership. Participants also emphasized the expectation that nurses should nurture patients’ strength, abilities, and determination to fight for their own lives.

The theme of *disempowerment* highlights participants’ experiences of reduced autonomy, compromised care quality, and perceived violations of their rights in the ICU. The most frequently cited concern was the lack of timely information about their condition, care plan, and prognosis. While it is essential for care providers to address this, simply delivering information in a clear manner is insufficient to support meaningful self-management. True empowerment requires engaging patients in decision-making, fostering understanding, and respecting their agency throughout the care process [48]. Patients also felt disappointed and disempowered when their request for physical care assistance, such as toileting, were delayed [19,21,26,33,38]. This study confirms that impersonal treatment, feelings of insecurity, and dependence on healthcare staff are significant sources of anxiety for ICU patients [49]. While healthcare teams focus on supporting survival, they may unintentionally undermine patients’ mental, physical, and psychological agency when patients feel overly dependent on them [12,24]. The desire for independence [21,25] is often coupled with feelings of being misunderstood [42] and unable to confide in others [42]. Sheridan et al. (2015) suggest that such feelings may stem from low engagement with care providers [48]. Similarly, Christensen and Hewitt (2007) highlight that fostering engaged nurse–patient relationships—such as regularly providing updates and information—can empower ICU patients [50]. Collectively, these studies emphasize the need to move beyond medical paternalism and prioritize patient empowerment by involving them in care decisions and pathways, ultimately fostering greater autonomy in their care.

The theme of *altered self-identity* suggests that critical illness and the process of care threaten patients’ identities. This is mostly a result of their loss of functional capacity, vulnerability, loss of dignity, cognitive changes, and influx of negative emotions, including delirium, while in the ICU. In line with this, many patients felt that technology controlled their physical existence, deepening their sense of lost autonomy. A phenomenological study similarly found that dependency on medical technology threatened patients’ sense of self, leading to a struggle for independence [51]. Our analysis also revealed that patients feared potential technology failures while relying on these systems. Supporting these findings, a meta-synthesis of 24 qualitative studies highlighted the importance of helping patients regain a sense of control by involving them in treatment decisions throughout their recovery process [52].

The theme of *torment* reflects participants’ awareness of the overwhelming control the ICU care system exerts over their bodies and personal wishes. Participants described ICU practices as mentally torturous, eroding their sense of safety and well-being. This theme highlights the restrictive and inflexible spatio-temporal dynamics of the ICU as central to patients’ lived experiences. Other studies revealed similar mental health experiences of patients in the ICU [9,53].

The finding that participants used words such as “prison” and “hell” to describe their care experiences in the ICU, aligns with Foucault’s (2003) notion on the inter-relations between medical knowledge and power [54]. In The Birth of the Clinic, Foucault (2003) describes that medical knowledge and power are interlinked and are applied to patients in a systematic manner [54]. This practice enables medical practitioners to validate and reproduce medical knowledge through their observations of patients’ bodies, and in essence, strengthen the healthcare provider’s power exercised on patients through medical knowledge. We extend Foucault’s argument to include that spatio-temporal restrictions, the criticality of illness, and practices of the ICU care also introduce another power imbalance between patients and health care providers. When this power is recognized, interrogated, and shared between patients and care providers, patients feel dignified, respected, supported, and empowered to pursue healing. On the other hand, when care providers focus solely on implementing care routines, and healthcare providers’ identified healthcare goals for patients, caring may, paradoxically, become oppressive and torturous to patients. Oppression from intubation and extubation represented the largest source of torture among patients [19,29,40,42] and were countered either with psychological resistance (fighting the ventilator) or acquiescence (accepting to submit to the will of the healthcare team) [12,19,25,29,42]. Our study calls on ICU care leaders, practitioners, and stakeholders to identify and implement patient-centered approaches for equalizing power with patients during their ICU care.

The theme of a *hostile environment* arose from participants’ perception of the ICU environment as hostile and traumatic. Participants described the ICU sounds and lights as tormenting. Moreover, participants perceived technology as a constant source of anxiety and fear. In line with our data, a recent systematic review revealed medical equipment, such as alarms inducing excess of noise, tubes and cables constraining the patient, bright lights, and sounds of other patients crying out as the highest sources of environmental stressors in the ICU [55]. A study using a sound location system revealed that the majority of loud sounds originated from physiological monitors and ventilators, sited close to patients’ ears [56]. Further analysis showed that the frequency range of the patient monitor alarms in the ICU (2.5–3.15 kHz) is similar to a human scream/baby’s cry. Although this may be ideal for attracting the attention of staff members, it is far from ideal for patients’ rest and comfort. Most of the ICU environmental stressors can be minimized by providing an atmosphere in which rest is possible [57]. For instance, conversations not directly involving patients or their visitors should occur away from the patient bedside. There is promising evidence on a sound reduction bundle comprising staff education, visual warnings of excessive sound levels, and monitor alarm reconfigurations [58]. Studies demonstrate that most alarms require no immediate action, instead precipitating alarm fatigue, leading to an increased risk of missing critical alarms and events [56,59]. Therefore, alarms on physiological monitors, ventilators, and other devices should be adjusted as required by the nursing staff.

Finally, our review also suggests that patients associate the criticality of the condition with impending death. *Fear of death* is a significant psychological stressor for patients in the ICU. This fear can impact not only the patients, but also their families and healthcare providers [60]. Understanding the dynamics of fear and its impact is essential for enhancing patient care and outcomes in the ICU. However, questions persist about the most effective ways to address patients’ fear of death [26]. Previous research found this fear to be more common among younger females [34], though our studies did not reveal a similar association. Nonetheless, evidence suggests that women may be more sensitive to emotional stimuli [61]. To better support patients facing this fear, nurses could be trained to deliver gender-tailored psychological interventions [12].

## 5. Practical Implications for Nursing and Healthcare Team

Given the critical nature of the ICU and the intense psychological burdens often experienced by patients, it is essential for nursing and healthcare professionals to adopt strategies that address both physical and emotional needs. The following practical implications are key to improving patient outcomes and fostering a more supportive ICU environment:Enhancing Patient-Centered CareNurses and healthcare professionals must prioritize patient autonomy and dignity, even in case of patients with impaired consciousness. This can be achieved through shared decision-making with patients and families, ensuring that care aligns with the patient’s values and preferences. Clear communications can help to mitigate feelings of powerlessness and perceived violations of autonomy. Moreover, empowering patients by involving them in their care/treatment pathways—whenever possible—can support their sense of control and mitigates feelings of helplessness.Integrating Psychological and Emotional Support into ICU CultureAddressing the psychological distress of ICU patients requires a fundamental shift in ICU culture, where emotional well-being is prioritized alongside implementing care routines for survival. Nurses should implement trauma-informed care strategies, therapeutic communication, and environmental modifications to foster a sense of safety and reduce perceptions of the ICU as a hostile or traumatic space. Routine psychological assessments should be incorporated into patient care, and healthcare teams should receive regular training in patient-centered communication and psychological resilience to ensure that ICU practices minimize distress and promote holistic well-being.Integrating Family for Emotional SupportEngaging family members as active participants in care can provide patients with a crucial sense of familiarity and reassurance, thereby alleviating psychological distress. For conscious patients, structured family interactions can help maintain emotional connections and enhance their sense of security. For unconscious patients, the presence of family members, along with verbal and non-verbal communication strategies, may contribute to emotional well-being and aid in recovery. Healthcare professionals should facilitate family engagement through flexible visitation policies and psychosocial support programs.

## 6. Limitations

Limiting the included studies to published English articles may introduce selection bias and influence the conclusions of this review. The authors chose English-language studies to ensure consistent interpretation and synthesis across the team. Additionally, the inclusion of studies published over a 21-year span may introduce recency bias when extrapolating lessons from the available evidence. However, it is important to note that significant changes in critical care policies and treatment protocols over the past two decades likely altered patient experiences, underscoring the need to explore patients’ perspectives within current care practices.

A potential limitation of the present meta-synthesis may be the fact we did not include patient-specific clinical data, e.g., reasons for ICU admission, length of stay, mental status, pain levels, hospital characteristics, and discharge details. Although such clinical data might provide additional context, they were not reported across the identified studies. Additionally, such analysis falls outside the scope of qualitative meta-synthesis. The focus of the present work was to synthesize patients’ lived experiences of psychological distress during ICU hospitalization. Thus, the present meta-ethnography focused on interpreting and integrating qualitative findings rather than extracting or correlating quantitative measures.

Similarly, although exploring how patients’ subjective distress correlates with objective clinical indicators may be of high clinical interest, empirical qualitative studies do not typically report quantitative measures. As a result, such correlations cannot be extracted from our dataset. Nevertheless, the present synthesis did not aim to establish causal relationships or predictive factors for psychological distress, but rather to capture the depth and complexity of patients’ experiences during ICU hospitalization. Yet, future research employing mixed-methods designs could provide insights into potential associations between patient experiences and clinical variables.

Many studies also showed gender imbalances, often including more male than female participants. Furthermore, most studies did not report racial or ethnic data, limiting the ability to analyze these factors. This highlights the need for future research to explore ethnic and gender-specific perceptions to better identify and address the psychological needs of ICU patients. Additionally, our synthesis incorporates studies from 19 different countries with varying ICU contexts—encompassing differences in ICU design, nurse-to-patient ratios, and institutional protocols and treatment strategies—and it does not explicitly account for ICU-specific contextual factors. However, existing evidence indicates that regardless of structural and procedural variations, the psychological experiences of ICU patients are globally consistent.

## 7. Conclusions

ICU admission imposes a significant psychological burden, often leaving a lasting emotional impact. The central theme of this evidence synthesis is the profound sense of disempowerment and loss of control experienced by patients. This pervasive feeling of powerlessness shapes how patients recount their experiences of surviving critical illness and ICU admission. Our review highlights patients’ strong desire for involvement, empathy, and empowerment throughout their care. To address this, future care standards should incorporate stress assessment and management strategies that prioritize patient empowerment and emotional safety during the care process. Additionally, it is imperative for healthcare professionals to prioritize patient autonomy and integrate psychological well-being into ICU care alongside physiological management. Future research should investigate the effects of patient empowerment and trauma-informed care on the short-term and long-term psychological health of ICU patients. Overall, the present findings contribute to understanding patients’ perspectives and emotional experiences.

## Figures and Tables

**Figure 1 healthcare-13-00894-f001:**
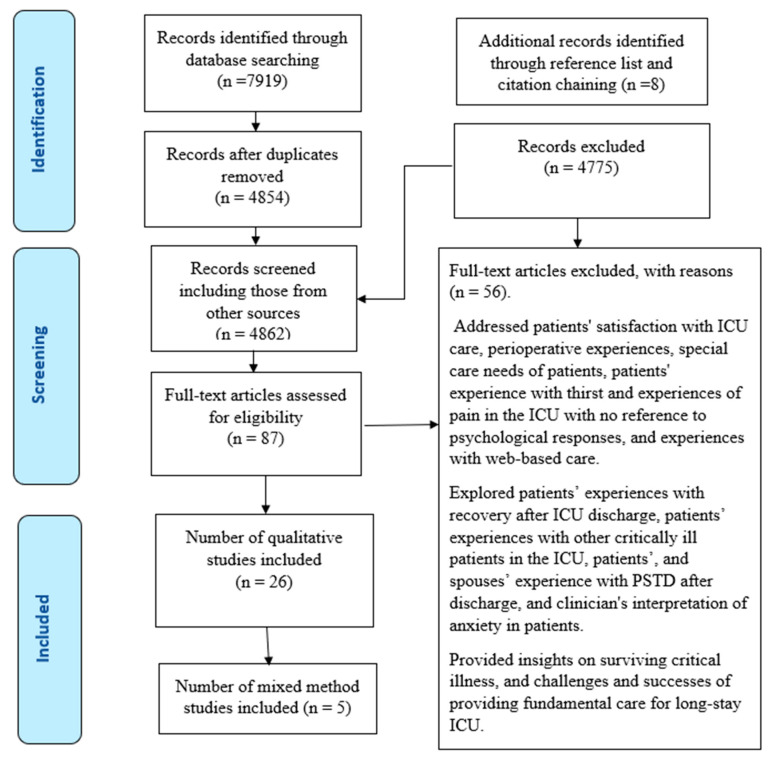
PRISMA chart for meta-ethnographyreview.

**Figure 2 healthcare-13-00894-f002:**
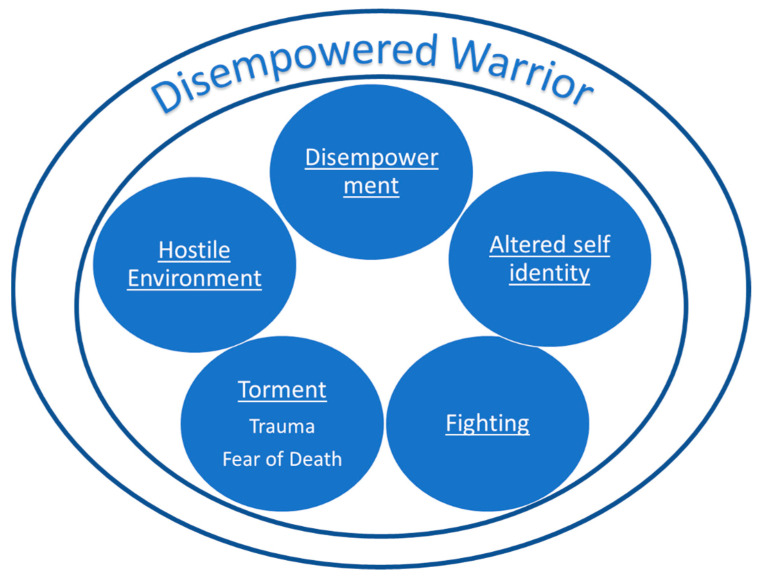
Emerging themes and synthesis.

**Table 1 healthcare-13-00894-t001:** Research question according to the SPIDER format.

**S**ample	Adult patients hospitalized or discharged from the ICU, who experienced psychological distress during ICU hospitalization.
**P**henomenon of **I**nterest	Psychological distress during ICU hospitalization. Psychological distress is defined as a subjective experience of discomfort and stress, which may also include a sense of demoralization, the experience of feeling broken or mental pain, a sense of anguish, symptoms of somatization and cognitive alterations, feelings of anger, self-perceived lack of control, and self-criticism [16].
Study **D**esign	Any type of qualitative research design (i.e., phenomenology, grounded theory, and ethnography) utilizing any type of qualitative data collection (i.e., interviews, focus groups, journals, and field notes).
**E**valuation	Experiences and/or perspectives and/or interpretations of psychological distress in the ICU and its impact on individuals’ perceptions and meanings of self and of their disease.
**R**esearch type	All types of published qualitative research papers and accessible grey literature. Mixed method studies with rich qualitative data were also included.

## Data Availability

The data supporting the findings of this study are available within the article.

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
