# Peer review of "Disempowered Warriors: Insights on Psychological Responses of ICU Patients Through a Meta-Ethnography"

_healthcare, 2025, doi:10.3390/healthcare13080894_

Round 1
Reviewer 1 Report
Comments and Suggestions for Authors
The article was conducted with a systematic review using a meta-ethnography method to understand the psychological reactions of patients treated in the intensive care unit (ICU) and to establish better care standards in this area.
1) Why is the web of science not included in the databases scanned? Some articles included in Pubmed are not included in the web of science. This may cause a selection bias.
2) The quality scores of the reviewed articles should be listed, and the number of points above should be stated in the evaluation.
A total of 31 studies from 19 countries were synthesized. What are the differences in ICU between countries? Information should be provided even if it is not among the main outcomes of the study.
One of the strengths of the article is the strong presentation of the findings using patient quotes.
If the points mentioned above are explained, I believe it can be published after minor revisions.
Author Response
Reviewer #1: Comments and Suggestions for Authors
The article was conducted with a systematic review using a meta-ethnography method to understand the psychological reactions of patients treated in the intensive care unit (ICU) and to establish better care standards in this area.
Response: Dear Reviewer, thank you for your thorough review of our manuscript. We greatly value your insights and the feedback you have provided. Below, we have addressed your concerns in detail.
Comment 1: Why is the Web of Science not included in the databases scanned? Some articles included in Pubmed are not included in the Web of Science. This may cause a selection bias.
Response: Although PubMed was not searched, Medline was included (these databases share the same Core database, Medline). Medline is preferred to PubMed due to the more advanced search interface (OVID), which greatly enhances the transparency and reproducibility of the search process. Several bibliographic databases with a variety of disciplinary focuses and generous indexes were searched, including Medline, EMBASE, PsycINFO, CINAHL, Scopus, and Dissertations and Theses Global - in combination, millions of journals were searched. We believe this grouping of databases greatly minimizes selection and geographic biases. The reason Web of Science was not included was because of the considerable overlap in journal indexing and, therefore, content between Scopus and Web of Science for a topic such as this one. So, working with a librarian well versed in expert searching and search strategies for synthesis methodologies, we determined searching both databases was not necessary.
Comment 2: The quality scores of the reviewed articles should be listed, and the number of points above should be stated in the evaluation.
Response: The CASP tool does not provide a scoring system, so instead of assigning points, we evaluated the quality as high, moderate, or low based on our appraisal. Please refer to table 3 page 11-13.
Comment 3: A total of 31 studies from 19 countries were synthesized. What are the differences in ICUs between countries? Information should be provided even if it is not among the main outcomes of the study.
Response: Thank you for your comment. The included studies varied in terms of ICU characteristics across different countries, including factors such as ICU design, nurse-to-patient ratios, and the standardized ICU protocols. We recognize the importance of contextualizing these differences. However, we were unable to summarize any key ICU differences due to the lack of information in the included studies. We have added this to the result section (page 5 line 142-145) and study limitation section (page 22 line 523-528).
Comment 4: One of the strengths of the article is the strong presentation of the findings using patient quotes. If the points mentioned above are explained, I believe it can be published after minor revisions.
Response: We greatly appreciate your feedback and hope that these revisions and explanations address your concerns.
Reviewer 2 Report
Comments and Suggestions for Authors
Dear Authors,
Congratulations on the topic, as it is innovative and relevant. Here are some suggestions for improvement.
The abstract is clear and adequately structured, highlighting the objectives, methods, and main findings.
The introduction adequately presents the context, clearly identifying the existing gap in the literature. However, it briefly lacks a deeper explanation regarding why a specific qualitative approach (meta-ethnography) was chosen. I suggest briefly elaborating on why a specific qualitative approach (meta-ethnography) was selected.
The methods section is robust but could be enhanced by explicitly stating the criteria used to determine data saturation in the synthesis. Additionally, the role of reviewers and the resolution process for specific disagreements should be described more clearly.
The results are comprehensive and clearly organized, with well-defined and structured categories. However, they are excessively detailed. It is suggested to reduce Table 2 to a more concise version, placing the detailed version in an appendix. Also, highlighting the most clinically relevant findings could improve practical applicability.
The discussion is well-developed but could deepen the relationship between the identified themes and practical implications for nursing and multidisciplinary teams in ICUs. For example, it could further explore how healthcare teams might apply these findings to modify practices and environments.
Finally, the conclusions are well-presented but superficial regarding how recommendations (empathy, control, empowerment) could be operationalized. Suggestions for future research that could address identified gaps should be included.
Best regards,
Author Response
Reviewer #2: Comments and Suggestions for Authors
Dear Authors,
Congratulations on the topic, as it is innovative and relevant. Here are some suggestions for improvement.
Response: Dear Reviewer, we would like to express our gratitude for the time and effort you have invested in reviewing our manuscript. Your thoughtful feedback has significantly contributed to improving our work. We have addressed your comments and concerns below.
Comment 1: The abstract is clear and adequately structured, highlighting the objectives, methods, and main findings.
Response: Thank you so much
Comment 2: The introduction adequately presents the context, clearly identifying the existing gap in the literature. However, it briefly lacks a deeper explanation regarding why a specific qualitative approach (meta-ethnography) was chosen. I suggest briefly elaborating on why a specific qualitative approach (meta-ethnography) was selected.
Response: Thank you for your insightful comment. Meta-ethnography was selected due to its interpretive nature and the facilitation of translation of concepts across studies. We have incorporated a brief description of this rationale in the manuscript; please refer to page 2 line 64-70.
Comment 3: The methods section is robust but could be enhanced by explicitly stating the criteria used to determine data saturation in the synthesis. Additionally, the role of reviewers and the resolution process for specific disagreements should be described more clearly.
Response: Thank you for your comment. In meta-ethnography, the concept of saturation is adapted to the context of synthesizing existing research rather than collecting new data. In our study, saturation was used to ensure conceptual clarity, that is, that the synthesis addressed the research questions adequately, represented the breadth and depth of available research data, while additional analysis did not contribute to the emerging themes. Please refer to page 4 line 126-131. We have also acknowledged the importance of clearly outlining the roles of the authors and the process for resolving disagreements. We added a description to the manuscript. Please refer to page 3 line 117-119.
Comment 4: The results are comprehensive and clearly organized, with well-defined and structured categories. However, they are excessively detailed. It is suggested to reduce Table 2 to a more concise version, placing the detailed version in an appendix. Also, highlighting the most clinically relevant findings could improve practical applicability.
Response: Thank you for your comment. We have revised table 2 accordingly; please refer to pages 5-10.
Comment 5: The discussion is well-developed but could deepen the relationship between the identified themes and practical implications for nursing and multidisciplinary teams in ICUs. For example, it could further explore how healthcare teams might apply these findings to modify practices and environments.
Response: Thank you for the comment. We have linked the themes with care practices throughout the discussion section page 20-22 (highlighted yellow), and also added practical implications for nursing and health care professionals to the manuscript. Please refer to page 22 line 466-492.
Comment 6: Finally, the conclusions are well-presented but superficial regarding how recommendations (empathy, control, empowerment) could be operationalized. Suggestions for future research that could address identified gaps should be included.
Response: Thank you for the comment. We have added a statement on directions for future research. Please refer to page 23 line 537-542.
Reviewer 3 Report
Comments and Suggestions for Authors
Dear authors,
I have thoroughly enjoyed reading your work and appreciate your insightful exploration of critical patients' perceptions of hospitalization in high-impact medical areas with intensive treatments and high mortality rates. The psychological impact on patients is significant, further complicated by interactions with medical staff.
I believe several aspects warrant further analysis, improvement, and clarification:
1. Firstly, please finalize and elaborate the study objectives, including a detailed secondary objective. Data analysis from the literature review will be completed by refining the analysis of included patients, specifically addressing: a) reasons for ICU admission (e.g., elective surgery, chronic decompensated diseases, polytrauma, acute cardiac events, oncological surgery); b) ICU length of stay; c) mental status (GCS) and pain level (using a standard pain scale) evolution during ICU stay; d) hospital level and profile details; e) inter-hospital protocol variations regarding patient relative access and video communication; and f) post-ICU transfer locations and patient independence levels at discharge.
2. Please provide a detailed analysis of the correlation between patient-reported perceptions of disease severity and objective clinical data.
3. Please reconsider the conclusions based on these findings.
Author Response
Reviewer #3: Comments and Suggestions for Author
Dear authors,
I have thoroughly enjoyed reading your work and appreciate your insightful exploration of critical patients' perceptions of hospitalization in high-impact medical areas with intensive treatments and high mortality rates. The psychological impact on patients is significant, further complicated by interactions with medical staff.
I believe several aspects warrant further analysis, improvement, and clarification.
Response: Dear Reviewer, we sincerely appreciate your time and effort in reviewing our manuscript. Your feedback has been valuable in helping us refine our work. Below, we address your concerns in detail.
Comment 1: Firstly, please finalize and elaborate the study objectives, including a detailed secondary objective. Data analysis from the literature review will be completed by refining the analysis of included patients, specifically addressing: a) reasons for ICU admission (e.g., elective surgery, chronic decompensated diseases, polytrauma, acute cardiac events, oncological surgery); b) ICU length of stay; c) mental status (GCS) and pain level (using a standard pain scale) evolution during ICU stay; d) hospital level and profile details; e) inter-hospital protocol variations regarding patient relative access and video communication; and f) post-ICU transfer locations and patient independence levels at discharge.
Response: We thank the reviewer for this comment, which helped us to enrich the section on limitations. Specifically, we acknowledge your suggestion to expand our study objectives to include a secondary objective that incorporates patient-specific clinical data (e.g., reasons for ICU admission, length of stay, mental status, pain levels, hospital characteristics, and discharge details). However, the studies included in our synthesis primarily reported narrative data on patient experiences rather than structured clinical variables such as GCS scores, pain scales, or discharge status. Consequently, it was not methodologically feasible to conduct an analysis of these factors within this meta-synthesis.
Additionally, we would like to clarify that our study is a qualitative meta-synthesis and, as such, does not incorporate or analyze quantitative clinical parameters. Specifically, the primary aim of our study is to synthesize patients' lived experiences of psychological distress during ICU hospitalization. Our methodology—meta-ethnography—focuses on interpreting and integrating qualitative findings rather than extracting or correlating quantitative measures.
Yet, to address this concern, we have added a brief text in our limitations section (page 22 line 502-510), acknowledging that while such clinical data might provide additional context, they fall outside the scope of qualitative meta-synthesis and are more suited for a quantitative systematic review or meta-analysis.
Comment 2: Please provide a detailed analysis of the correlation between patient-reported perceptions of disease severity and objective clinical data.
Response: We thank the reviewer for this comment which helped us to enrich the section of limitations. Indeed, we understand the interest in exploring how patients’ subjective distress correlates with objective clinical indicators. However, empirical qualitative studies do not typically report quantitative measures such as disease severity scores or correlations with patient-reported perceptions. As a result, such correlations cannot be extracted from our dataset.
Thus, a section clarifying this issue has been added in the “limitations” section, reporting that our synthesis did not aim to establish causal relationships or predictive factors for psychological distress but rather to capture the depth and complexity of patients’ experiences during ICU hospitalization. We also included a text suggesting that future research employing mixed-methods designs could provide insights into potential associations between patient experiences and clinical variables. Please refer to page 22-23 line 511-518.
Comment 3: Please reconsider the conclusions based on these findings.
Response: We thank the reviewer for this comment, which helped us to enhance the Conclusion section. Specifically, in light of your feedback, we have refined our conclusions to ensure they remain firmly grounded in the qualitative nature of our study. Specifically, we added a text emphasizing that our findings contribute to understanding patients’ perspectives and emotional experiences rather than identifying risk factors or measuring the effect size of psychological distress. Please refer to page 23 line 541-542.
We greatly appreciate your thoughtful critique and hope that these revisions clarify our study’s focus and methodological boundaries.
Round 2
Reviewer 2 Report
Comments and Suggestions for Authors
Dear Authors,
This article accept in present form.
Reviewer 3 Report
Comments and Suggestions for Authors
Dear authors,
The work shows substantial improvement. While some sources of error remain, due to the heterogeneity of the studies and the inherent subjectivity of patient experiences, I have no further questions or requests.